# Atomic Simulations of the Interaction between a Dislocation Loop and Vacancy-Type Defects in Tungsten

Linyu Li [1], Hao Wang [2], Ke Xu [1], Bingchen Li [1], Shuo Jin [1], Xiao-Chun Li [3], Xiaolin Shu [1], Linyun Liang [1,*] and Guang-Hong Lu [1,*]

1    School of Physics, Beihang University, Beijing 100191, China; linyu@buaa.edu.cn (L.L.); xuuke@buaa.edu.cn (K.X.); by1919006@buaa.edu.cn (B.L.); jinshuo@buaa.edu.cn (S.J.); shuxlin@buaa.edu.cn (X.S.)
2    Center for Fusion Science, Southwestern Institute of Physics, Chengdu 610041, China; hwang@swip.ac.cn
3    Institute of Plasma Physics, Chinese Academy of Sciences, Hefei 230031, China; xcli@ipp.ac.cn
*    Correspondence: lyliang@buaa.edu.cn (L.L.); lgh@buaa.edu.cn (G.-H.L.)

**Abstract:** Tungsten (W) is considered to be the most promising plasma-facing material in fusion reactors. During their service, severe irradiation conditions create plenty of point defects in W, which can significantly degrade their performance. In this work, we first employ the molecular static simulations to investigate the interaction between a 1/2[111] dislocation loop and a vacancy-type defect including a vacancy, di-vacancy, and vacancy cluster in W. The distributions of the binding energies of a 1/2[111] interstitial and vacancy dislocation loop to a vacancy along different directions at 0 K are obtained, which are validated by using the elasticity theory. The calculated distributions of the binding energies of a 1/2[111] interstitial dislocation loop to a di-vacancy and a vacancy cluster, showing a similar behavior to the case of a vacancy. Furthermore, we use the molecular dynamics simulation to study the effect of a vacancy cluster on the mobility of the 1/2[111] interstitial dislocation loop. The interaction is closely related to the temperature and their relative positions. A vacancy cluster can attract the 1/2[111] interstitial dislocation loop and pin it at low temperatures. At high temperatures, the 1/2[111] interstitial dislocation loop can move randomly. These results will help us to understand the essence of the interaction behaviors between the dislocation loop and a vacancy-type defect and provide necessary parameters for mesoscopic scale simulations.

**Keywords:** atomic simulations; dislocation loop; vacancy defect; tungsten

## 1. Introduction

Tungsten (W), owing to its high melting temperature, good thermal conductivity, and low sputtering yield, is believed to be the most promising candidate for plasma-facing materials (PFMs) in fusion reactors [1–4]. During the operation of the reactors, high-energy neutrons escaped from the plasma will bombard on PFMs, creating plenty of self-interstitial atoms (SIAs) and vacancies in them [5]. These point defects further aggregate into small vacancy clusters including voids and dislocation loops [6–10]. Experimental observations showed voids and dislocation loops are major defect clusters in pure W at the low dpa level (less than 1.54) in the temperature range from 300 to 900 °C [11]. The dislocation loop can be either the vacancy dislocation loop or the interstitial dislocation loop with different properties [10–13]. The existence of various irradiated defects makes the system difficult to be understood. The interactions between them further complex the system, which is believed to play an important role in the microstructural evolution of PFMs. For examples, the dislocation loop can grow or shrink by absorbing SIAs and vacancies, respectively [14,15]. Two dislocation loops attract or repel each other when they are in different relative positions [16]. The dislocation loops and voids act as an obstacle to the dislocation glide [17–23]. For the interaction between the dislocation loop and the dislocation, it closely depends on the character and nature of the loop, and their relative

positions and temperature [16,18–20]. For the interaction between voids and dislocations, the presence of an edge dislocation in the vicinity of the void can generate a stress field that impacts on the motion of the dislocation [23]. Therefore, understanding the interaction between these defects is very important to explore their microscopic mechanism and thus correlate them to the macroscopic mechanical properties [10,16,20,21,23].

Several previous studies focused on the interaction behaviors between vacancy-type defects and dislocation loops in iron [23–26]. Dislocation loops act as biased sinks that attract SIAs and vacancies. For the interaction between a vacancy and SIA clusters (9-127SIAs), the vacancy can be annihilated only when it is placed along the edge of the dislocation loop and parallel to the Burgers vector. When the vacancy is in a site next to the center of the cluster, it does not annihilate with SIAs but affects the motion of the cluster, reducing or even preventing its migration [24]. Furthermore, their interactions are temperature and cluster size dependent. Thus, the vacancy has an influence on the movement and evolution of the dislocation loop. When the vacancies aggregate into vacancy clusters, they can further interact with the dislocation loop in a large distance. The vacancy clusters can attract the dislocation loop due to the elastic interaction between them [27]. Molecular dynamics (MD) simulations showed that when the vacancy cluster is placed within the interaction distance to a 1/2[111] dislocation loop in W, it directly diffuses towards the vacancy cluster. The diffusion speed of the dislocation loop is related to the shape, size, and position of the vacancy clusters. The vacancy clusters can be annihilated eventually if the dislocation loop is large enough [27]. However, although several studies of the interaction between vacancy-defects and dislocation loops have been done for iron, there is still a lack of relevant investigations on W, in which the stable structure of the interstitial crowdion and the diffusion mechanism of the dislocation loop are different from that of ion [28,29]. Therefore, the objective of this work is to systematically investigate the interaction between a dislocation loop and a vacancy-type defect in W.

In this work, we study the interaction behaviors of the dislocation loop and a vacancy, di-vacancy, and vacancy clusters by using the molecular statics (MS) and MD simulations. We first calculate the binding energies of two types of 1/2[111] dislocation loops to a monovacancy. To validate the simulation results, the widely-used elasticity theory (ET) is performed to calculate their binding energies [24–26]. Then we calculate the binding energies of the 1/2[111] interstitial dislocation loop to a divacancy and a vacancy cluster using MS simulations. The effect of the vacancy cluster on the mobility of the 1/2[111] dislocation loop at different temperatures is also investigated using MD simulations. We hope our results can provide useful datasets for large-scale simulations such as kinetic Monte Carlo, cluster dynamics, and dislocation dynamics and help to study the long-term and large-scale microstructure evolution in W under irradiation.

## 2. Simulation Details

We employ the Large-scale Atomic/Molecular Massively Parallel Simulator (LAMMPS) software (Sandia National Laboratories, Albuquerque, NM, USA) to study the interaction between a dislocation loop and vacancy-type defects including a vacancy, di-vacancy, and vacancy cluster. A previous study indicated that nearly 60% of the dislocation loop is the 1/2[111] dislocation loop in W at 0.4–30 dpa and temperatures between 300 and 750 °C [21]. The size of the dislocation loop is usually less than 20 nm with most of them less than 6 nm [22]. Thus, we choose the 1/2[111] dislocation loop with its radius less than 6 nm. The embedded atom method (EAM) empirical interatomic potential developed by Marinica et al., denoted as "EAM2", is employed in our simulations, which has been used to investigate the properties of dislocation loop [30,31].

The direction of X, Y, and Z is set as $[\bar{1}\bar{1}2]$, $[1\bar{1}0]$, and $[111]$, respectively. The simulation box created by LAMMPS contains about 0.59 million atoms. Periodic boundary conditions are imposed on all boundaries of the simulation box. The 1/2[111] dislocation loop is placed at the center of the box and the vacancy-type defects are created at designed positions. The

sketch of the simulation box and a schematic picture of the configuration of a 1/2[111] dislocation loop and a vacancy-type defect is shown in Figure 1.

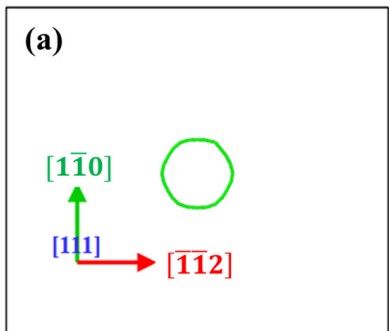 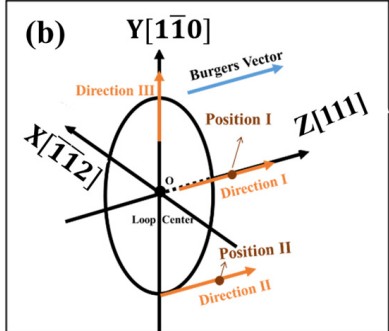

**Figure 1.** (**a**) The sketch of the simulation box. The green ring represents the 1/2[111] dislocation loop. (**b**) The schematic picture of the configuration of a 1/2[111] dislocation loop and a vacancy-type defect. The vacancy-type defect is placed at the position that is along three different directions. Direction I is through the center of the dislocation loop and parallel to the Burgers vector. Position I is along the Direction I and has a distance of 15 Å to the habit plane (HP). Direction II is along the edge of the dislocation loop and parallel to the Burgers vector. Position II is along the Direction II and has the distance of 15 Å to the HP. Direction III is through the center of the dislocation loop on the HP.

Binding energies can be used to evaluate the static interaction between a dislocation loop and a vacancy-type defect. We first construct a 1/2[111] dislocation loop at the center of the simulation box. Then we insert a vacancy-type defect at a certain position (Position I and II) as shown in Figure 1b and relax the system to reach the equilibrium state by using the conjugate gradient method. The binding energy of the loop to a vacancy-type defect is given by:

$$E_b = E_1 - E_2 \tag{1}$$

where $E_1$ stands for the minimum energy of the system with a vacancy-type defect that is far away from the dislocation loop to ensure that there is no interaction between them, and $E_2$ stands for the minimum energy of the system with a vacancy-type defect that is in a specific position. A positive $E_b$ represents the attraction of the dislocation loop to a vacancy-type defect, while the negative value denotes the repulsion between them.

The elasticity theory is first used to verify our MS simulation results. A vacancy can be regarded as a sphere whose elastic constants are zero. On the one hand, the elastic constant of a vacancy is different from that of system, which will interfere with the existing elastic field and produce the interaction energy $E_{int}^1$. On the other hand, the stress produced by the vacancy will interact with the original elastic field and produce the interaction energy $E_{int}^2$. According to the linear elasticity theory, the total interaction energy is equal to the sum of the two energies $E_{int}^1$ and $E_{int}^2$ [32]. Then the interaction energy between a dislocation loop and a vacancy can be obtained. The HP of the dislocation loop is $(r, \theta)$ with a radius $R$. The center of the loop is at the origin and the Burgers vector is parallel to the Z axis. The interaction energy between a vacancy and a loop can be calculated by the following formula [32]:

$$E_{int}^1(\xi, \rho) = -K^2(1-\sigma)\Omega\left\{\frac{1}{3\kappa}\frac{(1+\sigma)^2}{1-2\sigma}I_0^{1\,2} + \frac{15}{2\mu}\frac{1}{7-5\sigma}\left[\frac{(1-2\sigma)^2}{3}\left(I_0^1\right)^2 + \xi^2 I_0^{2\,2} + \xi^2 I_1^{2\,2} + \rho^{-2}\Phi^2 + \xi\rho^{-1}I_0^2\Phi - (1-2\sigma)\rho^{-1}I_0^1\Phi\right]\right\}, \tag{2}$$

$$E_{int}^2(\xi, \rho) = -\frac{2}{3}K(1+\sigma)\Delta\Omega I_0^1 \tag{3}$$

where:

$$\xi = \frac{z}{R} \tag{4}$$

$$\rho = \frac{r}{R} \tag{5}$$

$$K = \frac{b}{R} \frac{\mu}{2(1-\sigma)} \tag{6}$$

$$I_m^n(\xi,\rho) = \int_0^\infty t^n J_m(t\rho) J_1(t) e^{-\xi t} dt \, (n, m = 0, 1, 2) \tag{7}$$

$$\Phi(\xi,\rho) = (1-2\sigma)I_0^1 - \xi I_1^1 \tag{8}$$

where $(z, r)$ is the position of the vacancy, $R$ is the dislocation loop radius, $k$ and $\mu$ represents the elastic constant tensor $C_{12}$ and $C_{44}$ of W, respectively. $k_1$ and $\mu_1$ represent the elastic constant of the vacancy, which equals 0. $\sigma$ is the Poisson's ratio. $\Omega$ is the volume of the vacancy, and $I_m^n(\xi,\rho)$ is a complete elliptic integral. The used elastic constant tensors of W are $C_{11}$ = 523 GPa, $C_{12}$ = 203 GPa, and $C_{44}$ = 160 Gpa.

## 3. Results and Discussion

We first use the MS to calculate the binding energies of the 1/2[111] interstitial dislocation loop (IDL) and 1/2[111] vacancy dislocation loop (VDL) to a vacancy by varying the type and size of the loop at 0 K. The binding energies of the IDL to a vacancy are compared with that calculated by ET. We then simulate the binding energies of the IDL to a di-vacancy and a vacancy cluster. Finally, the effects of the temperature and position of the vacancy cluster on the mobility of the IDL are obtained by the MD simulations.

### 3.1. Interaction between an 1/2[111] IDL to a Vacancy

Two types of the 1/2[111] dislocation loop, IDL and VDL, can be experimentally observed [21]. To understand the interaction mechanism of the IDL and VDL to a vacancy, we calculate their binding energies by varying the relative position of the vacancy. Figure 2 depicts the binding energies of the IDL to a vacancy on the HP of the loop along the Direction I and Direction II.

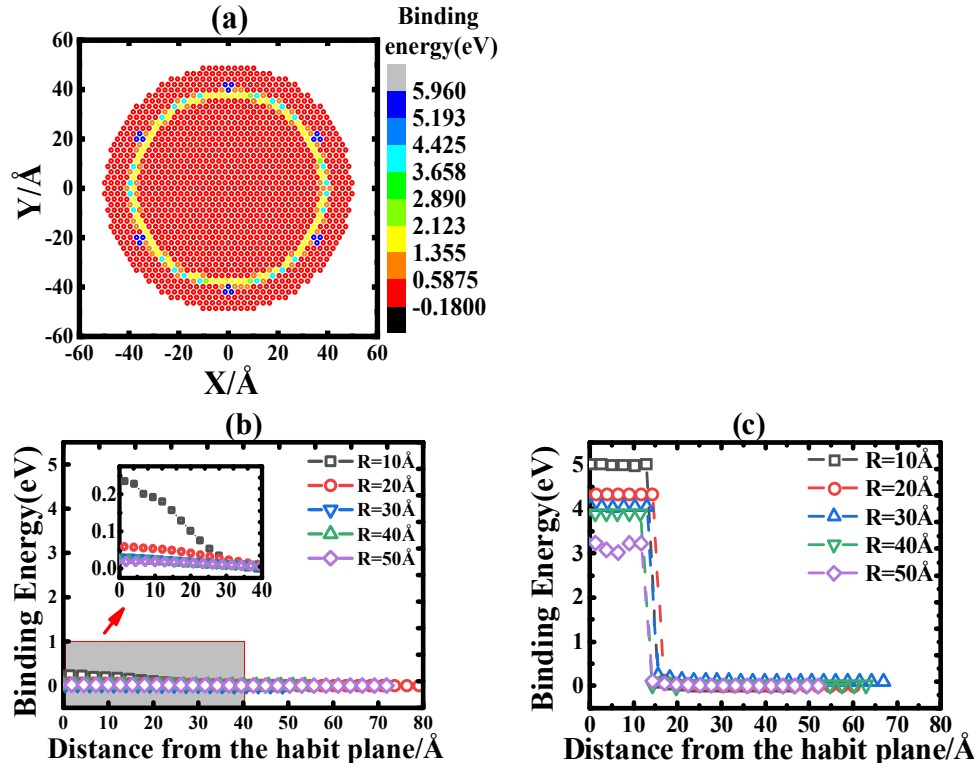

**Figure 2.** The distribution of binding energies of an IDL to a vacancy when the vacancy is placed on the HP (**a**), along the Direction I (**b**), and along the Direction II (**c**), respectively. The inset figure shows the binding energies at a small energy scale.

The distribution of the binding energies of IDL (R = 40.0 Å) to a vacancy placed on the HP (XY plane) is shown in Figure 2a. We choose the center of the loop as the origin of the coordinate. The binding energy is calculated by creating a vacancy at every lattice position on the HP. The color bar indicates the values of the binding energies of the IDL to the vacancy. It can be seen that the binding energy is relatively small when the vacancy is placed far away from the dislocation loop. The binding energy becomes large when the vacancy is placed close to the edge of the IDL, where the vacancy can be absorbed. We define the regime that the IDL can absorb the vacancy as their absorption area. This absorption area has a ring-like shape and its width is defined as the absorption distance. The calculated absorption distance is ~8.0 Å. The binding energies approach zero when the vacancy is not in the absorption area.

Due to the symmetrical distribution of the binding energies on the HP as shown in Figure 2a, we calculate the binding energies only along one particular direction to save computational resources. Figure 2b shows the calculated distribution of binding energies of the IDL to a vacancy as a function of their distance along the Direction I. With the increase of the distance, the binding energies gradually decrease for the same sized IDL. The binding energies are almost zero (<0.025 eV) when the distance is larger than 30.0 Å for all sized IDL. As the radius of the IDL is increased from 10.0 Å to 50.0 Å, the highest binding energy decreases from 0.25 eV to 0.01 eV. Figure 2c shows the distribution of the binding energies between the IDL and vacancy as a function of their distance along the Direction II. The binding energies remain constants within a certain distance for the same sized IDL. This indicates the vacancy is absorbed by IDL. Beyond this distance, the binding energies are nearly zero (<0.15 eV). This distance is defined as the absorption distance along the Direction II. With the radius of the IDL increasing from 10.0 Å to 50.0 Å, the largest binding energy decreases from 5.1 eV to 1.4 eV and the absorption distance is 14.0–17.0 Å. There is no clear evidence that the IDL size has an influence on the absorption distance based on our simulation results.

Comparing the binding energies calculated along the Direction I and Direction II, we can find that the binding energies along the Direction II are always larger than that along the Direction I for the same sized IDL with the same relative distance. This implies that the attraction of the IDL to the vacancy is stronger along the Direction II than that of the Direction I. The vacancy can be annihilated by IDL only when it is placed close to the edge of the loop.

We find that a stable VDL cannot be formed in the system if the radius of the VDL is smaller than 40.0 Å in our simulations. It will evolve into a void. Therefore, we construct a series of VDLs with a radius of 40.0 Å to 60.0 Å. Figure 3 shows the distribution of the binding energies of VDL to a vacancy on the HP of the loop along the Direction I and Direction II, respectively.

The distribution of the calculated binding energies of the VDL (40.0 Å) to a vacancy placed on the HP is shown in Figure 3a. It shows the absorption area of the VDL to a vacancy also has a ring-like shape, which is very similar to the case of the IDL to a vacancy. The largest binding energy is around 2.2 eV, which is smaller than that of the IDL to a vacancy. The binding energies approach to zero when the vacancy is placed outside the absorption area.

Figure 3b depicts the distribution of the binding energies as a function of their distance along the Direction I. The binding energies are negative for all sized VDL, implying the repulsive interaction between the VDL and the vacancy. The largest binding energy increases from −0.032 eV to −0.02 eV with the radius of the VDL increasing from 40.0 Å to 60.0 Å. As the distance increases, the value of the binding energy increases for the same sized VDLs. If the distance is larger than 30.0 Å, the value of the binding energy is less than 0.015 eV. The distribution of the binding energies as a function of their distance along the Direction II is shown in Figure 3c. The largest binding energy decreases from 2.4 eV to 0.9 eV with the increase of the radius of the loop from 40.0 Å to 60.0 Å. Within the absorption distance (~15.0 Å), the binding energy of a VDL to a vacancy decreases

gradually that is different from the case of the IDL and a vacancy. Beyond this absorption distance, the binding energies are nearly zero.

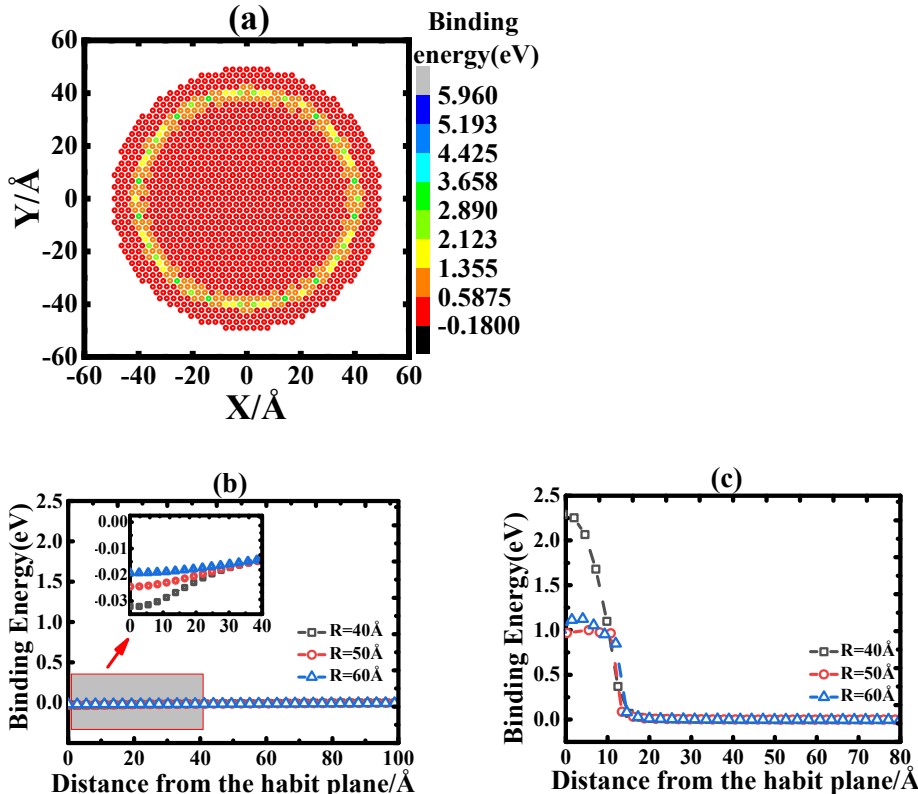

**Figure 3.** The distribution of binding energies of a VDL to a vacancy when the vacancy is placed on (**a**) HP, along (**b**) the Direction I and (**c**) Direction II, respectively. The inset figure shows the binding energies at a small energy scale.

Based on the above simulation results, the distribution of the binding energies of a VDL to a vacancy are similar to that of an IDL to a vacancy. However, we find that the binding energies of the IDL to a vacancy are positive while the binding energies of the VDL to a vacancy are negative. Thus, the IDL can attract the vacancy while the VDL can slightly repulse it along the Direction I. The reason can be explained by the distribution of the stress for different dislocation loops. As shown in Figure 4a, there is a compressive stress inside the IDL but a tensile stress outside it. While the signs of the stress are opposite for the VDL as shown in Figure 4b. A vacancy is very easily combined with the IDL, which is the main reason to have positive binding energies inside the IDL. The different distribution of the stress contributes to the different interaction behaviors of the IDL and VDL to a vacancy.

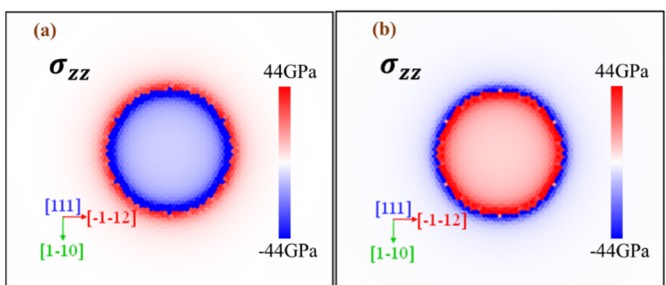

**Figure 4.** The stress distributions of the (**a**) interstitial dislocation loop and (**b**) vacancy dislocation loop with a radius of 40.0 Å.

### 3.2. Comparison of the Binding Energies of IDL-Vacancy by Using ET and MS

To better understand the interaction behaviors between a vacancy and a 1/2[111] IDL, we compare their binding energies with different relative positions calculated by using both MS and ET as shown in Figure 5. The radius of the IDL is chosen as 40.0 Å as an example. The vacancy is placed at several selected positions along the Direction II and Direction III.

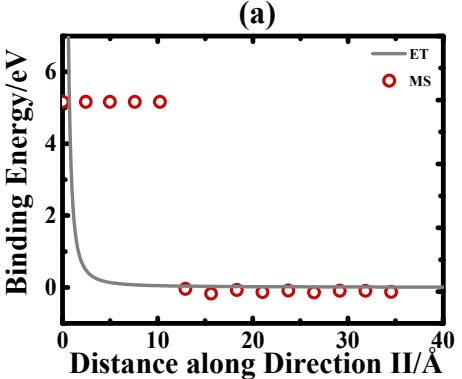 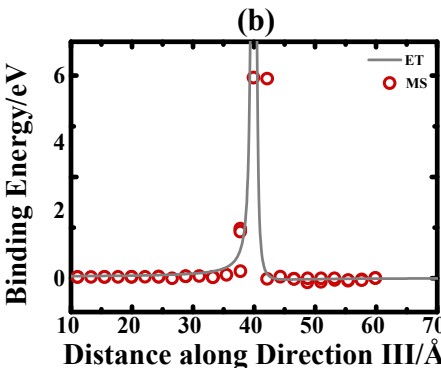

**Figure 5.** Comparison of the binding energies of the IDL (40.0 Å) to a vacancy calculated by MS and ET along (**a**) the Direction II and (**b**) the Direction III, respectively.

Figure 5a shows the comparison of the binding energies between an IDL and a vacancy calculated by ET and MS as a function of their distance along the Direction II. When the distance is less than 14.0 Å, the binding energies calculated by ET decrease dramatically from an infinity value to zero with increasing the distance, while the binding energies calculated by MS are nearly a constant. The difference can be attributed to the ET cannot accurately describe the stress field when two defects are close to each other, and also the vacancy is completely absorbed by IDL in MS simulations that is not the case in ET. Beyond this distance, the binding energies given by ET and MS are in good agreements. Figure 5b shows the comparison of the binding energies calculated by the ET and MS as a function of their distance along the Direction III. Based on the difference of the binding energies between the IDL and a vacancy, we can categorize the distribution of the binding energies into two different regimes, inside the absorption distance that is greater than 35.0 Å and less than 42.0 Å and beyond the absorption distance. Within the absorption distance, the binding energies predicted by ET tend to be infinity, while the binding energy given by MS is 5.93 eV. Beyond the absorption distance, the binding energies calculated by ET agree well with that given by MS. Furthermore, the stress inside the loop is larger than that outside the loop as shown in Figure 4a. Thus, it can be seen from Figure 5b that the binding energies given by ET are slightly larger for the vacancy placed inside the loop than that outside the loop along the Direction III. In conclusion, we basically validate our MS simulation results although the ET can not fully describe the interaction between an IDL and a vacancy.

### 3.3. Static Interaction of an IDL to a Di-Vacancy and a Vacancy Cluster

To investigate the interaction between the IDL to a di-vacancy and a vacancy cluster, we calculate their binding energies along the Direction II and Direction III, respectively. The configuration of the divacancy considered is 1NN, which was reported stable in W [6]. Figure 6 shows the distribution of the calculated binding energies between the different sized IDLs and a di-vacancy as a function of their distance along the Direction I and II, respectively. The variations of the binding energies with the distance and loop radius are very similar to that of an IDL and a vacancy as shown in Figure 2. The largest binding energy of the IDL to a divacancy is generally larger than that of the IDL to a vacancy. Along the Direction I as shown in Figure 6a the largest binding energy decreases from 0.45 eV to 0.03 eV with increasing the radius of the IDL from 10.0 Å to 50.0 Å. The binding energy reaches zero when the distance is larger than 40.0 Å for all sized IDLs. While along the

Direction II as shown in Figure 6b, the largest binding energy decreases from 5.8 eV to 1.2 eV with increasing the radius of the loop from 10.0 Å to 50.0 Å. The calculated absorption distance ranges from 15.0 Å to 27.0 Å.

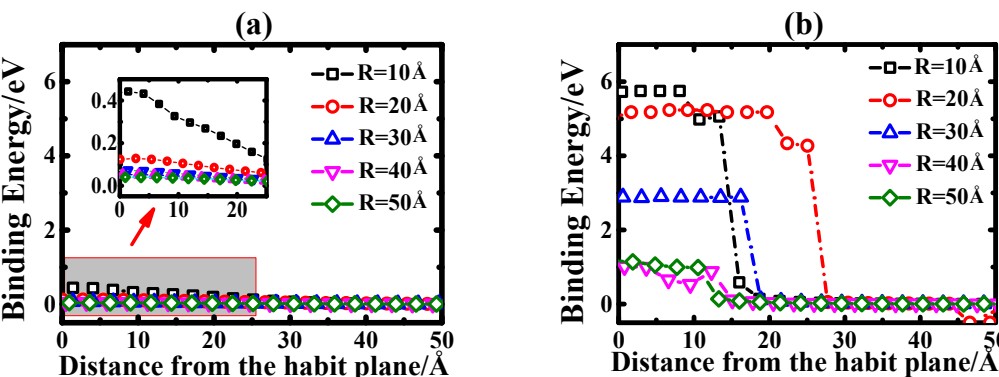

**Figure 6.** The distribution of binding energies of the different sized IDLs to a divacancy as a function of their distance when the divacancy is placed along (**a**) the Direction I and (**b**) Direction II, respectively. The inset figure shows the binding energies at a small energy scale.

Figure 7 depicts the calculated binding energies between the different sized IDLs and a vacancy cluster as a function of their distance along the Direction I and II. Comparing to the binding energies of the IDL to a vacancy and a di-vacancy as shown in Figures 5 and 6, the distribution of the binding energies of IDL to a vacancy cluster as a function of their distance shows a similar trend but larger values. Figure 7a shows the binding energy decreases with the increase of their distance and the loop radius along the Direction I. The largest binding energy decreases from 3.4 eV to 0.05 eV with increasing the radius of the loop from 10.0 Å to 50.0 Å. The binding energy is almost zero when the distance is longer than 40.0 Å for all sized IDLs. While along the Direction II as shown in Figure 7b, the largest binding energy decreases from 12.0 eV to 6.2 eV as the IDL radius increases from 10.0 Å to 50.0 Å. The absorption distance ranges from 22.0 Å to 32.0 Å. Therefore, as the number of vacancies increases, the binding energies increase gradually for the same sized IDL and the same distance. Besides, the distribution of the binding energies of the IDL to various vacancy-type defects as a function of the distance along the Direction I and II is similar. Based on the same trend of the distribution of the binding energies between the IDL and vacancy-type defects, we can conclude that they have similar interaction behaviors.

### 3.4. Dynamic Interaction between an IDL and a Vacancy Cluster

A previous study showed that the IDL exhibits a fast one-dimensional motion along the <111> direction in W[16]. The existence of a vacancy inside the IDL can inhibit its motion, which also closely depends on the IDL size and temperature [24]. Besides, the number of the vacancies can also have a large influence on the movement of the IDL or even absorb it [27]. Therefore, the vacancy cluster plays an important role in the evolution of the IDL and needs to be investigated systemically.

We calculate the movement distance of the IDL (10.0Å) along the Burgers vector with a vacancy cluster (containing eight vacancies) by varying its position and temperature. The movement distances of the IDL along the Burgers vector as a function of the simulation time at different temperatures are shown in Figure 8. The inset pictures show the movement distance of the IDL at the initial stage (0–150 ps). The interaction between the IDL and the vacancy cluster largely depends on the temperature.

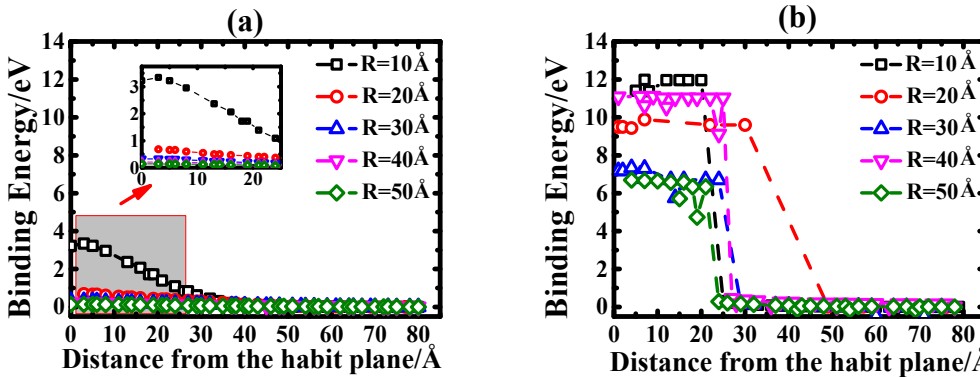

**Figure 7.** The distribution of binding energies of the different sized IDLs to a vacancy cluster (containing eight vacancies) as a function of their distance when the vacancy clusters is placed along the (**a**) Direction I and (**b**) Direction II, respectively. The inset figure shows the binding energies at a small energy scale.

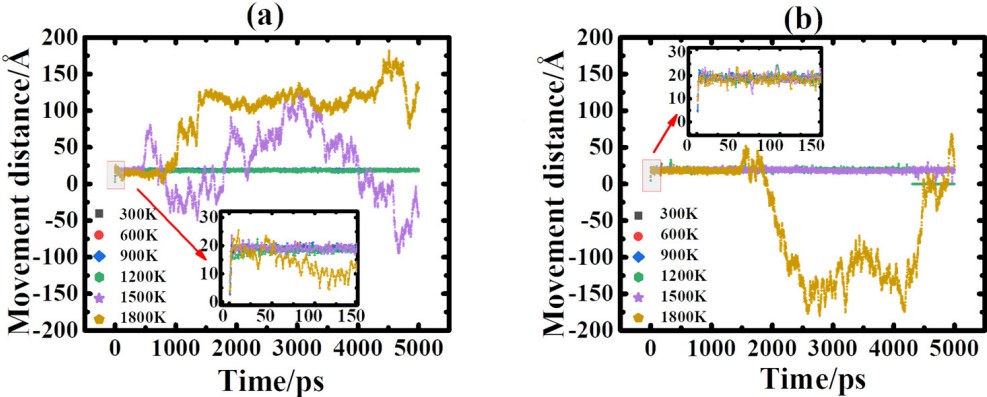

**Figure 8.** The movement distance of the IDL along the Burgers vector when a vacancy cluster is placed at (**a**) Position I and (**b**) Position II, respectively.

Figure 8a shows the movement distance of the IDL with the vacancy cluster placed at the position I as a function of the simulation time at different temperatures. We can see that the IDL will diffuse towards the vacancy cluster at the initial stage as shown in the insert figure. Then the IDL is captured by the vacancy cluster for all temperatures. At a temperature below 1200 K, the simulation results show the IDL is pinned by the vacancy cluster and immobile. At a temperature above 1500 K, the IDL can move after 400 ps and 800 ps, respectively. Based on the analysis of the evolution of the microstructure, the vacancy cluster first attracts the IDL to drive it to the position of the vacancy cluster, then the vacancies inside the cluster will be absorbed by IDL. However, not all vacancies in the vacancy cluster will be absorbed. The left vacancies will inhibit the movement of the IDL. At a temperature above 1200 K, the vacancy captured inside the IDL can migrate towards the edge of the IDL, and then the vacancy can be annihilated. Finally, the loop mobility is recovered. Figure 8b shows the calculated movement distance of the IDL with a vacancy cluster placed at Position II at several selected times. It is shown that the interaction behavior is very similar to that with a vacancy cluster placed at Position I. While the IDL requires a higher temperature of 1800K than that at Position I to unpin the vacancy cluster.

In conclusion, the movement distance of the IDL depends on the position of the vacancy cluster and temperature. At low temperature, the IDL is pinned by the vacancy cluster, while it can move at high temperature because it can promote the recombination of the IDL and the vacancy. Besides, the attracting capability of the IDL by the vacancy cluster placed at Position II is stronger than it placed at Position I.

## 4. Conclusions

In this work, we employ an atomic simulation to investigate the interaction behaviors between a dislocation loop and vacancy-type defects including a vacancy, di-vacancy, and vacancy cluster. The distribution of the binding energies of an 1/2[111] interstitial dislocation loop (IDL) to a vacancy is different from that of a 1/2[111] vacancy dislocation loop (VDL) to a vacancy. When the vacancy is adjacent to the center of the loop, the VDL repulses the vacancy, but the IDL attracts it due to their different stress distributions. The binding energies calculated by elasticity theory (ET) and molecular statics simulations show that they are consistent when they are far away from each other, but it has a large derivation when they are close due to the ET has a difficulty in accurately predicting the stress near the core of the loop. The interaction behaviors between the IDL and a di-vacancy are very similar to that of the IDL and a vacancy. Furthermore, a vacancy cluster (containing 8 vacancies) can hinder the motion of the IDL, which depends on the temperature and their relative position. No matter where the vacancy cluster is, the IDL absorbs a part of the vacancies in the cluster and the unabsorbed vacancies will inhibit the motion of the dislocation loop at low temperatures. While the mobility of IDL is recovered at high temperature by absorbing all vacancies in nanoseconds. Therefore, the temperature and position dependent interaction of the IDL and a vacancy cluster should be taken into account in modeling the microstructure evolution during irradiation. These obtained binding energies and absorption distances provide input parameters for the kinetic Monte Carlo, cluster dynamics, and dislocation dynamics simulations.

**Author Contributions:** Conceptualization, G.-H.L., L.L. (Linyun Liang), H.W., K.X. and L.L. (Linyu Li); methodology, G.-H.L., L.L. (Linyun Liang), X.-C.L., H.W., K.X. and L.L.(Linyu Li); software, L.L. (Linyu Li); validation, L.L. (Linyu Li).; formal analysis, G.-H.L., L.L.(Linyun Liang), X.-C.L., H.W., K.X., B.L., S.J., X.S. and L.L. (Linyu Li); software, L.L. (Linyu Li); investigation, L.L. (Linyu Li); resources, L.L. (Linyu Li); data curation, L.L. (Linyu Li); writing—original draft preparation, L.L. (Linyu Li); writing—review and editing, G.-H.L., L.L.(Linyun Liang), X.-C.L., H.W., K.X., B.L., S.J., X.S. and L.L.(Linyu Li); visualization, L.L. (Linyu Li). All authors have read and agreed to the published version of the manuscript.

**Funding:** This research was funded by the National Natural Science Foundation of China, Grant Numbers. 51871007, 12075021, and 12075023, and the National MCF Energy R&D Program of China, Grant Number No. 2018YFE0308103. The APC was funded by Grant Number No. 2018YFE0308103.

**Data Availability Statement:** The data presented in this study are available on request from the corresponding author. The data are not publicly available due to technical limitations.

**Acknowledgments:** The authors acknowledge Xinyue Fan for providing inspiration and dealing with technical problems.

**Conflicts of Interest:** The authors declare no conflict of interest.

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
