# Peer review of "Atomic Simulations of the Interaction between a Dislocation Loop and Vacancy-Type Defects in Tungsten"

_metals, doi:10.3390/met12030368_

Round 1

Reviewer 1 Report

Analysis of  interactions between vacancy clusters (voids) and dislocation loops in tungsten (W) is very important to realize  microscopic mechanisms producing the macroscopic mechanical properties of tungsten under high-energy irradiation. The reviewed manuscript is one of the steps in this direction. I have a couple of comments on the content of this overall good work article.

The authors' use of the abbreviation MS for the term Molecular Dynamic is somewhat unclear. It is introduced in line 67 of the article. What is it done for? Why can't the authors use the abbreviation MD, as it is introduced in line 57 ?

The originality of formulas 1 and 2 is also not quite clear. If the formulas have been derived earlier, the authors should insert references to a textbook on the elasticity theory  or to the original work containing a derivation of these formulas.

In principle, the present paper can be published after a minor revision.

Reviewer 2 Report

In this work, the authors have used molecular static simulations and molecular dynamics to investigate the interaction between 1/2[111] dislocation loop and a vacancy-type defect, including a vacancy, di-vacancy, and vacancy cluster in W. The work is of great importance and yielded interesting results, helping to understand better the nature of the interaction between a dislocation loop and a vacancy defect as a function of distance and temperature in W.

However, the authors should consider the following points before consideration of publication.

(1) In Fig1, the simulation box should be illustrated in addition to the schematic picture.

(2) E1 is the minimum energy of the system with a vacancy-type defect far away from the dislocation loop to ensure no interaction between them. In this case, the dislocation-vacancy distance should be maximum equal to the half of the simulation box's lattice parameter; otherwise, if the distance is greater than the half, how much the energy E1 is affected by the interaction of the vacancy defect and the dislocation image created by the periodic boundary condition?!

(3) What is the fundamental difference between MS and MD methods in this work?

(4) Looking at the energy figures, at first glance, the reader can not realize the big difference between the binding energies of Fig.2b and c, for example. They look similar to each other, but in reality, those of Fig.2c are 20 times larger in order of magnitude. Therefore, the authors are invited to use the same scale for both the figures, with an additional inserted figure within Fig 2b, showing the energies at a small scale. The same applies for  Fig. 3b and c, Fig.4a and b, Fig.5a and b, Fig.6a and b, Fig.7a and b.

(5) The binding energies of an IDL to vacancy are positive, while those of VDL to vacancy are negative, creating IDL-vacancy attraction and VDL-vacancy repulsion. The authors explained that this is caused by the compressive stress induced inside the IDL and the tensile stress outside it, while the signs of the stress for the VDL are opposite. I agree, but the authors are invited to share stress values as evidence also, the effect of chemical interaction contribution, in that case, should be discussed.

Reviewer 3 Report

Dear authors,

thank you for you interesting and original work. Have a good works in the future.

Best regards,

reviewer

Round 2

Reviewer 2 Report

The authors have answered all my comments and cleared my questions.
I recommend the acceptance of the present form of the manuscript for publication in Metals after correction of the following minor points:
(1) In the caption of Fig.4, the (a) label is typed twice.
(2) In Fig.5, the (a) label is missing and should be added.
(3) A period should be marked after the numeral 4 in "4. Conclusions" instead of the comma in "4, Conclusions".
